# 3DSPA: A 3D Semantic Point Autoencoder for Evaluating Video Realism

## Abstract

AI video generation is evolving rapidly. For video generators to be useful for applications ranging from robotics to film-making, they must consistently produce realistic videos. However, evaluating the realism of generated videos remains a largely manual process – requiring human annotation or bespoke evaluation "datasets" which have restricted scope. Here we develop an automated evaluation framework for video realism which captures both semantics and coherent 3D structure and which does not require access to a reference video. Our method, 3DSPA is a 3D spatiotemporal point autoencoder which integrates 3D point trajectories, depth cues, and DINO semantic features into a unified representation for video evaluation. 3DSPA models how objects move and what is happening in the scene, enabling robust assessments of realism, temporal consistency, and physical plausibility. Experiments show that 3DSPA reliably identifies videos which violate physical laws, is more sensitive to motion artifacts, and aligns more closely with human judgments of video quality and realism across multiple datasets. Our results demonstrate that enriching trajectory-based representations with 3D semantics offers a stronger foundation for benchmarking generative video models, and implicitly captures physical rule violations.

## 1 Introduction

Recent years have witnessed rapid progress in generative video models, with systems such as Sora (Brooks et al. (2024)), Kling AI (Kuaishou Technology (2024)), and Luma-Ray (LumaAI (2025)) capable of producing high-resolution, long-duration videos conditioned on natural language descriptions. These systems have started to showcase unprecedented visual fidelity, with coherent multi-objects, smooth camera motion, and diverse scenes. However, the end objective of developing these text-to-video models has always been to generate videos which are not only visually compelling but also realistic—capturing semantic meaning, temporal consistency, and physical plausibility in a way that mirrors a real-world video. If achieved, it will generate tremendous excitement across domains ranging from robotics and embodied AI (Wu et al. (2023); Yang et al. (2025); Fu et al. (2025)) to virtual reality (Christian et al. (2025)), education (Xu et al. (2025)), and creative industries like advertising and film-making.

Understanding the realism of generated videos is more than an aesthetic problem, it directly affects their utility for various downstream applications. In robotics and embodied AI, for example, policies trained in simulated environments that fail to accurately capture real-world dynamics may not transfer successfully to deployment settings. Similarly, in entertainment, audiences are sensitive to subtle cues of unrealistic motion, which can undermine immersion. Thus, a systematic way of measuring whether generated videos are physically plausible and perceptually realistic is a foundational requirement for both scientific and practical use.

However, existing approaches to measuring realism remain limited. The most common strategy is to rely on human annotation, where raters provide subjective assessments of qualities such as naturalness, temporal

smoothness (Wu et al. (2021); Skinner et al. (2023)). While such annotations are informative, they are expensive, time-consuming, and do not scale to the vast number of videos modern generative systems can produce. A second line of work has attempted to build discriminative benchmarks by constructing datasets of paired real and fake videos (Borji (2022)), training classifiers to distinguish them. Yet this requires careful curation of datasets, often domain-specific, and assumes that generated samples are comparable to available real-world footage. Neither approach provides a scalable, general-purpose solution.

Moreover, prior automated measures have largely equated realism with temporal consistency—ensuring that videos do not exhibit frame-to-frame flickering or incoherence. While temporal smoothness is indeed important, it is not sufficient. Realism also requires adherence to the semantics of motion and the physical laws that govern objects in three dimensions. For example, a video where a ball bounces upward indefinitely without slowing down might look temporally smooth but is physically implausible. Likewise, a car turning a corner but sliding sideways without frictional constraints violates semantic expectations of how vehicles move. Prior work has struggled to capture such failures because they demand reasoning about both semantics and 3D structure, not just pixels over time. Most existing evaluations (Allen et al. (2025), operate in 2D feature spaces, neglecting the fact that real-world objects persist in three dimensions, maintain continuity across occlusion, and obey physical laws such as gravity, inertia, and collision.

To address these challenges, we propose **3DSPA** (3D Semantic Point Autoencoder), a novel framework for assessing the realism of generated videos. 3DSPA combines semantic features with 3D point track autoencoding. The key idea is to represent a video as a sequence of tracked 3D points, enriched with semantic embeddings, and train an autoencoder that reconstructs these tracks. By compressing and reconstructing motion trajectories, the model is forced to capture underlying physical and semantic regularities, making deviations from realism detectable.

The main contributions of our paper include -

- First, we demonstrate that 3DSPA functions as a capable 3D point tracker, despite the information bottleneck inherent in autoencoding.
- Second, we show that it reliably detects violations of physical laws in controlled synthetic settings by using the IntPhys2 benchmark (Bordes et al. (2025)).
- Finally, we evaluate 3DSPA on existing works of using human annotations to judge realism using two datasets of generated videos, EvalCrafter (Skinner et al. (2023)) and VideoPhy-2 (Bansal et al. (2025)), and find that it better aligns with human judgments of motion quality and physical realism compared to existing baselines.

Together, these results suggest that incorporating semantics and 3D structure is essential for scalable, automated evaluation of generative video realism.

## 2 RELATED WORK

**Physical Benchmarking for Video Models** Benchmarking intuitive and broader physical understanding has been central to recent progress in machine perception. Earlier works include IntPhys (Riochet et al. (2018)) evaluating models on core physical expectations through possible vs. impossible event videos. PHYRE (Bakhtin et al. (2019)) frames intuitive physics as counterfactual puzzle-solving, requiring agents to reason about object interactions. For broader reasoning, CLEVRER (Yi et al. (2020)) advances causal reasoning in videos via descriptive, explanatory, and counterfactual questions. More recently, Physion++ (Bear et al. (2023)) extends visual physics prediction tasks to richer scenarios involving rigid, soft, and fluid dynamics, providing a comprehensive evaluation suite. IntPhys2 (Bordes et al. (2025)) was recently released which is based on the violation of expectation framework challenge models to differentiate between possible and impossible events within controlled and diverse virtual environments.

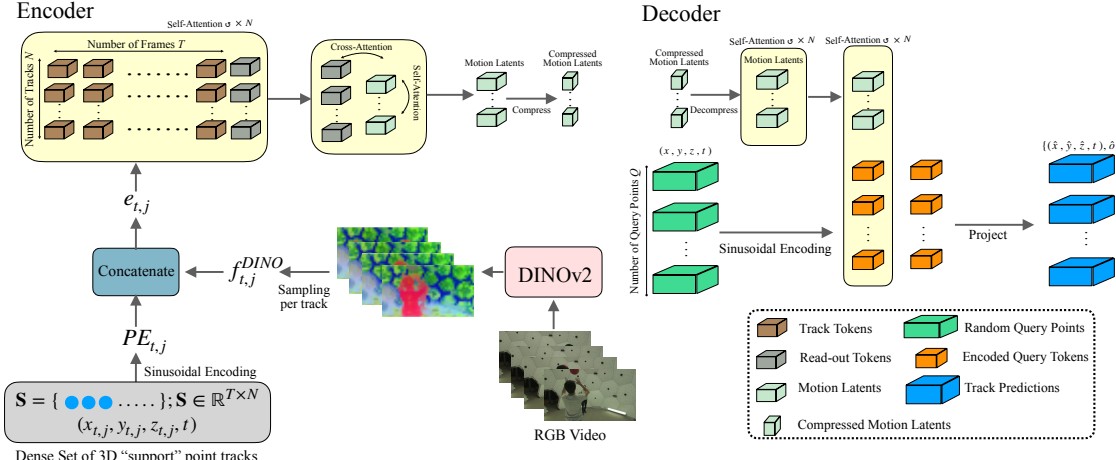

Figure 1: **3DSPA Architecture Overview :** The encoder integrates 3D trajectories, temporal embeddings, and DINOv2 ((Oquab et al., 2023)) semantic features into a compact latent representation using occlusion-aware attention and Perceiver-style transformer architecture (Jaegle et al. (2021)). The decoder conditions on query points to reconstruct full 3D trajectories with occlusion flags.

**Video Quality Assessment** Text-to-Video generative models are rapidly progressing, but it is still unclear how far they are from being able to generate videos which are indistinguishable from reality. Even evaluating this progress remains tricky. Earlier methods include FVD (Unterthiner et al. (2018)) and CLIP (Radford et al. (2021)) to evaluate the quality of frames and the text-frame alignment respectively. However, these metrics cannot capture realism more broadly, which simultaneously incorporates semantics and geometric structure. Recent works aim to create benchmarks with automated evaluators which tackle realism more directly, e.g. (Chen et al. (2025b)), and VBench (Huang et al. (2024)). However, many models have started saturating these benchmarks, achieving high scores of 90%+, since they are simply not challenging enough. Benchmarks such as EvalCrafter (Skinner et al. (2023)) and VideoPhy2 (Bansal et al., 2025) instead resort to human raters to perform comprehensive evaluation, and therefore avoid issues of benchmark saturation. Automated evaluators in these settings include optical flow and vision-language models, but no approach fully captures human assessments of motion quality and realism.

## 3 MODEL

Our goal is to provide an automated metric which can capture human ratings of realism for any video. To that end, we introduce 3DSPA : 3D Semantic Point Autoencoder. 3DSPA can be viewed as an extension of TRAJAN (Allen et al., 2025), a 2D point trajectory autoencoder that is trained to map a support set of point trajectories into a fixed-size motion latent representation which is further used to reconstruct query point tracks. While TRAJAN does a good job in capturing motion information in latents and reconstruction, it has some limitations. First, the model has no knowledge of the surrounding environment, only the point trajectories themselves. This means it cannot reason about scene context, object interactions, or occlusions that can be crucial for judging motion plausibility. Second, restricting motion to 2D trajectories is not sufficient for evaluating realistic dynamics, which naturally occur in 3D. As a result, TRAJAN cannot fully

capture the complexity of real-world motion. 3DSPA is instead designed to reconstruct 3D point tracks from random queries across space and time and provide semantic-aware motion latent representation.

## 3.1 ARCHITECTURE

3DSPA adopts an encoder–decoder setup, where the encoder $E$ operates on a dense set of support point tracks $\mathbf{S} = \{s_{t,j}\}$, with each track defined as $s_{t,j} = (x_{t,j}, y_{t,j}, z_{t,j}, o_{t,j})$. Here, $(x, y, z)$ denote the 3D position and $o$ is a binary occlusion flag at time $t$ for the $j$-th track. The model is trained to reconstruct a separate set of query trajectories $\mathbf{Q} = q_{t,j}$, which are randomly sampled from the video.

For each trajectory $j$, we embed its 3D positions $(x_{t,j}, y_{t,j}, z_{t,j})$ together with time $t$ using sinusoidal encoding (denoted by $\text{PE}_{t,j}$). In parallel, we sample DINOv2 (Oquab et al. (2023)) embeddings $f_{t,j}^{\text{DINO}}$ from the corresponding video frame regions. These two representations are concatenated as $e_{t,j} = [\text{PE}_{t,j} \,\|\, f_{t,j}^{\text{DINO}}]$, and then projected into $C$ channels.

A learnable "readout" token is initialized randomly and appended, and self-attention is applied across all the tokens with an occlusion-aware mask $(1 - o_{t,j})$ to ignore hidden points. After attention, only the readout token is retained, producing a compact $C$-dimensional descriptor for the track. To integrate information across tracks, we adopt a Perceiver-style transformer (Jaegle et al. (2021)) where a set of 128 latent tokens cross-attend to all track descriptors and then interact through self-attention. Finally, the latent tokens are compressed to yield a fixed $128 \times 64$ representation $\phi_S$, capturing both motion dynamics and semantic appearance cues.

The decoder in 3DSPA follows the same design as TRAJAN, but now operates on a motion latent $\phi_S$ that already integrates 3D dynamics and semantic context. We train the decoder to reconstruct held-out query tracks. Concretely, given $\phi_S$ and a query point $(x_q, y_q, z_q, t_q)$, the decoder predicts the full trajectory passing through that point. We first up-project the latent tokens in $\phi_S$ with an operator $U$ and add a query *readout* token obtained from a sinusoidal encoding of $(x_q, y_q, z_q, t_q)$. Self-attention is applied over all tokens, after which only the readout token is retained. A final linear projection maps this token to the predicted trajectory $(\hat{x}_t^q, \hat{y}_t^q, \hat{z}_t^q, \hat{o}_t^q)$ across all frames.

### 3.1.1 TRAINING

Following CoTracker3 (Karaev et al. (2024)), we train 3DSPA on a combination of synthetic and real datasets to ensure both controlled supervision and real-world generalization. We use the Kubric3D dataset generator (Greff et al. (2022)) to create 38k synthetic scenes with ground-truth 3D trajectories. While Kubric3D directly provides $(x_{t,j}, y_{t,j}, z_{t,j})$ for every point $j$ at time $t$, it does not include explicit occlusion labels. To obtain occlusion flags, we project each 3D point into the image plane and compare its depth $z_{t,j}$ against the rendered depth map $D_t(x_{t,j}, y_{t,j})$ at that pixel. For depth maps we use VideoDepthAnything (VDA) metric model (Chen et al. (2025a)). The occlusion flag is then defined as $o_{t,j} = \mathbf{1}[\, z_{t,j} > D_t(x_{t,j}, y_{t,j}) + \epsilon \,]$, where $\epsilon$ is a small tolerance of 1e-4 to account for numerical precision. Thus, $o_{t,j} = 1$ indicates that the point is occluded at time $t$. In addition, we use the TAPVid-3D dataset Koppula et al. (2024), a large benchmark covering diverse real-world scenarios. TAPVid-3D contains 4,569 videos in the main split and 150 videos in the minival split, with lengths ranging from 25 to 300 frames. Unlike Kubric3D, TAPVid-3D provides full ground-truth annotations, including 3D point trajectories as well as occlusion flags $o_{t,j}$, making it directly suitable for evaluating models under realistic settings. In our setup, we train on the main set and use the minival split for evaluation. This benchmark also provides the necessary metrics for evaluation which we discuss later.

During training, we randomly divide the trajectories in each video into two equal halves. The first half is used as support tracks, which are passed through the encoder to produce the motion latents. The second half is used as query tracks, which the decoder must reconstruct given only the motion latents.

We trained 3DSPA with the AdamW Loshchilov & Hutter (2017) optimizer using a learning rate of 1e-4 warm-up followed by cosine decay. We initialize 3DSPA with a pretrained TRAJAN checkpoint and train for 300 epochs. Depth predictions are regularized with a scale-invariant penalty to handle differences in scale between synthetic and real domains. The DINO module (Oquab et al. (2023)) is frozen during training. Additional training details, including hyperparameter settings, batch sizes, and ablation studies, are provided in Appendix A.

The training loss is :

$$\mathcal{L}_{\text{3DSPA}} = \sum_{q,t} \Big( w_{l1} \|(x_{q,t}, y_{q,t}, z_{q,t}) - (\hat{x}_{q,t}, \hat{y}_{q,t}, \hat{z}_{q,t})\|_1 + w_{BCE}\text{BCE}(o_{q,t}, \hat{o}_{q,t}) \Big).$$

### 3.1.2 INFERENCE

At inference, we operate directly on 2D input videos but we require 3D point tracks. Dense 2D point tracks $(x_{t,j}, y_{t,j})$ with occlusion are first estimated using CoTracker3 Karaev et al. (2024), and subsequently lifted into 3D with metric depth predictions from VideoDepthAnything (VDA) metric model (Chen et al. (2025a)). The resulting 3D tracks $(x_{t,j}, y_{t,j}, z_{t,j})$ are then provided to the trained model in a 1:1 ratio as both support and query tracks. The reconstructed tracks produced by the decoder are finally used for evaluation.

Specifically, we calculate the Average Jaccard ($AJ_{3D}$) of the reconstructed tracks as a quantitative metric to see reconstruction error. Following TAPVid-3D (Koppula et al. (2024)), as AJ increases, the quality of reconstruction increase and vice-versa. The AJ metric calculates the number of true positives (number of points within the $\delta_{3D}$ threshold, predicted correctly to be visible), divided by the sum of true positives and false positives (predicted visible, but are occluded or farther than the threshold) and false negatives (visible points, predicted occluded or predicted to exceed the threshold).

## 4 RESULTS

To demonstrate that 3DSPA can capture realistic, physical motion, we evaluate three complementary axes: its accuracy in 3D point tracking as described in Section 4.1, its ability to detect physical law violations in possible vs. impossible video pairs as described in Section 4.2, and its alignment with human judgments of realism in generated videos as described in Section 4.3.

### 4.1 CAN 3DSPA RECONSTRUCT 3D POINT TRACKS?

We evaluate 3DSPA on the TAPVid-3D minival set and report three 3D point tracking metrics: Occlusion Accuracy (OA), which measures the precision of occlusion predictions; $APD_{3D}$, the average percentage of errors within multiple threshold scales $\delta$; and Average Jaccard ($AJ_{3D}$), which quantifies the accuracy of both position and occlusion estimation. All these metrics are taken from Koppula et al. (2024)'s work.

Since 3DSPA is an *autoencoder* of point tracks, and therefore inherently less accurate due to its information bottleneck, we do not expect its performance in 3D point tracking to rival state-of-the-art approaches. Nevertheless, it is important that 3DSPA can reasonably accurately reconstruct 3D point tracks. We therefore compare 3DSPA against 3D-lifted versions of state-of-the-art 2D tracking methods and 3D tracking methods like SpatialTracker (Xiao et al. (2024) and SpatialTrackerv2 (Xiao et al. (2025)). Since most of these models were originally trained on the synthetic Kubric3D (Greff et al. (2022)), while our training data combines both Kubric3D (synthetic) and TAPVid-3D (real), we additionally fine-tune CoTracker3 model (Karaev et al. (2024)) on TAPVid-3D pseudo labels and evaluate all models on the minival set. Table 1 summarizes the comparative 3D tracking performance. 3DSPA consistently outperforms most baselines and

| Method | Aria | | | DriveTrack | | | PStudio | | | Average | | |
|---|---|---|---|---|---|---|---|---|---|---|---|---|
| | AJ ↑ | APD ↑ | OA ↑ | AJ ↑ | APD ↑ | OA ↑ | AJ ↑ | APD ↑ | OA ↑ | AJ ↑ | APD ↑ | OA ↑ |
| TAPIR + ZD | 15.7 | 23.5 | 79.8 | 6.3 | 10.5 | 81.6 | 11.2 | 18.9 | 78.7 | 11.0 | 17.6 | 80.1 |
| CoTracker + ZD | 17.0 | 25.7 | 88.0 | 6.0 | 10.9 | 82.6 | 11.4 | 19.9 | 80.0 | 11.4 | 18.8 | 83.5 |
| BootsTAPIR + ZD | 11.8 | 16.3 | 86.7 | 6.4 | 10.9 | 85.3 | 11.6 | 19.6 | 82.6 | 11.6 | 18.9 | 84.9 |
| CoTracker3 + ZD | 15.6 | 24.1 | 88.6 | 13.3 | 19.6 | 86.8 | 9.0 | 13.6 | 83.9 | 12.6 | 19.1 | 86.4 |
| SpatialTracker | 16.7 | 25.7 | 89.3 | 6.9 | 12.4 | 83.7 | 12.3 | 21.6 | 78.5 | 12.0 | 19.9 | 83.8 |
| SpatialTrackerV2 | 18.6 | 26.3 | 90.8 | 16.4 | 24.3 | 90.2 | 18.1 | 27.6 | 86.7 | 17.7 | 26.0 | 89.2 |
| CoTracker3-FT + ZD | 16.8 | 25.5 | 89.6 | 13.6 | 19.9 | 88.7 | 10.1 | 14.3 | 87.1 | 13.5 | 19.9 | 88.5 |
| Ours | 17.7 | 24.9 | 89.2 | 11.9 | 14.8 | 85.7 | 12.3 | 19.9 | 82.5 | 14.0 | 19.8 | 85.8 |

Table 1: 3D point tracking results on the TapVid-3D *minival* set. 3DSPA achieves competitive accuracy across datasets and performs on par with a finetuned CoTracker3 (CoTracker3-FT+ZD), highlighting its ability to reconstruct consistent and accurate 3D tracks.

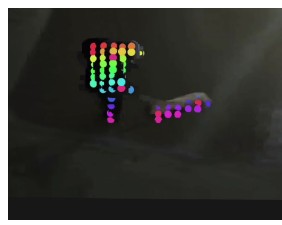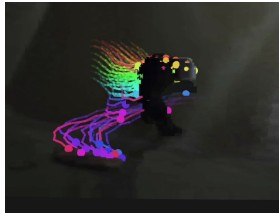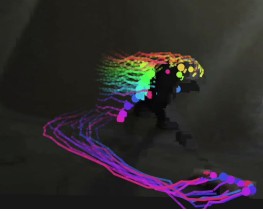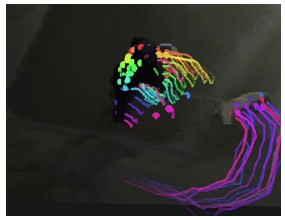

Figure 2: Example 3D point tracks reconstructed by 3DSPA for a generated video of a man mopping a floor in the VideoPhy-2 dataset (Bansal et al., 2025). This video has a good human rating and was reconstructed pretty well by our model.

achieves performance on par with CoTracker3 (Karaev et al. (2024)) when fine-tuned on the TAPVid-3D main dataset.

We additionally provide an example of how 3DSPA performs in 3D track reconstruction when only a 2D input video is provided in Figure 2. Despite the noisy depth signal obtained from VideoDepthAnything, 3DSPA reconstructs smooth 3D tracks for the generated video.

Both results demonstrate that 3DSPA is capable of reconstructing 3D point tracks accurately despite its compressed latent space bottleneck, and motivates its candidacy as an automated metric for video realism.

### 4.2 CAN 3DSPA DETECT PHYSICAL RULE VIOLATIONS?

For an automated metric of video realism to be useful, we need to be sure that it will detect physical rule violations. To assess whether 3DSPA can reliably distinguish physically real and unreal scenarios, we evaluate on the IntPhys2 (Bordes et al. (2025)) dataset. IntPhys2 contains 1,012 videos across 253 scenes, organized as quadruplets of two **possible** (real) and two **impossible** (unreal) outcomes. Each video tests one of four core physical principles: **object permanence**, where objects continue to exist even when occluded; **object immutability**, where objects maintain their shape and structure; **spatio-temporal continuity**, where objects move smoothly through time and space; and **solidity**, where objects occupy space and cannot pass through one another. All videos are rendered in the Unreal Engine with both static and moving cameras, increasing realism and memory demands.

**Baselines and ablations** We compare 3DSPA against several state-of-the-art vision-language models, self-supervised vision foundation models, and TRAJAN variants that progressively add dimensional and seman-

| Model / Category | Permanence | | Immutability | | Continuity | | Solidity | |
|---|---|---|---|---|---|---|---|---|
| | Fixed | Moving | Fixed | Moving | Fixed | Moving | Fixed | Moving |
| GPT-4o (Hurst et al. (2024)) | 59.62 | 58.82 | 58.65 | 59.56 | 54.81 | 57.35 | 56.73 | 55.32 |
| Qwen-VL 2.5 (Bai et al. (2025)) | 53.85 | 54.41 | 56.73 | 53.68 | 52.88 | 54.41 | 50.96 | 51.06 |
| Gemini-1.5 Pro (Google DeepMind (2024)) | 55.77 | 55.88 | 56.73 | 56.73 | 54.80 | 54.80 | 56.73 | 56.73 |
| Gemini-2.5 Flash (Google DeepMind (2025)) | 64.42 | 58.82 | 59.62 | 63.97 | 54.81 | 55.15 | 55.77 | 56.38 |
| VideoMAEv2-g (Wang et al. (2023)) | 63.46 | 50.00 | 54.81 | 53.69 | 65.38 | 54.41 | 48.08 | 59.57 |
| Cosmos-4B (Agarwal et al. (2025)) | 51.92 | 41.18 | 50.96 | 48.32 | 53.85 | 50.00 | 48.08 | 55.32 |
| V-JEPA 2-h (Assran et al. (2025)) | 63.46 | 67.65 | 51.92 | 56.38 | 50.00 | 57.35 | 50.00 | 52.13 |
| V-JEPA-h+RoPE (Bardes et al. (2024) ) | 59.62 | 57.35 | 55.77 | 58.72 | 57.69 | **75.00** | 46.15 | 58.51 |
| TRAJAN | 44.23 | 50.00 | 54.81 | 58.82 | 53.85 | 48.53 | 46.15 | 52.13 |
| TRAJAN+DINO | 61.54 | **76.47** | **75.00** | 73.08 | **78.85** | 73.53 | 69.23 | 59.57 |
| TRAJAN+3D | 65.38 | 60.29 | 46.15 | 66.18 | 50.00 | 58.82 | 38.46 | 39.36 |
| 3DSPA | **76.92** | 75.00 | 73.08 | **76.47** | 67.31 | 69.12 | **70.77** | **64.47** |
| Human | 100.0 | 99.26 | 97.11 | 90.44 | 99.04 | 94.44 | 96.15 | 95.21 |

Table 2: Win rates (%) on IntPhys2 across physical principles. Top row reports prior models' win rates. Bottom rows benchmark against 3DSPA , ablations, and human performance. 3DSPA and TRAJAN+DINO strongly outperform all alternatives in detecting physically implausible events across most concept categories.

tic information. The original TRAJAN model uses only 2D point tracks without depth cues or semantic features. **TRAJAN+3D** is a 3D extension where we add an extra spatial dimension in the autoencoder to better capture motion dynamics. **TRAJAN+DINO** instead augments the representation with semantic features from DINOv2, while still excluding 3D information and depth cues. Together, these variants highlight the individual roles of 3D structure and semantic context in detecting physical rule violations.

**Performance Analysis.** Table 2 shows the performance of 3DSPA against the baselines reported in Bordes et al. (2025) as well as our ablations. 3DSPA and TRAJAN+DINO significantly outperform all alternatives across most concept categories. Perhaps most surprisingly, 3DSPA shows the *most benefit* over alternatives in the permanence ( $+10\%$), immutability ( $+10\%$), and solidity ( $+5\%$) concept categories rather than continuity (approximately $-5$ to $+2\%$). This suggests that a small amount of 3D point track data is sufficient for models to learn what is physically plausible or not, and that reconstructing semantic 3D tracks is a better signal for learning *realistic, plausible* physical motion than next frame prediction or next token prediction.

Looking at the ablations, most of the benefits of 3DSPA in determining possible vs. impossible physics may be due to the inclusion of DINO features. Although 3DSPA performs best overall, TRAJAN+DINO performs comparably to 3DSPA in most concept categories, indicating that *semantic information* is key for understanding physical principles. By comparison, TRAJAN and TRAJAN+3D perform comparably to previously evaluated predictive and Multimodal LLM (MLLM) approaches. We provide additional results in Appendix B.

### 4.3 DOES 3DSPA CAPTURE HUMAN EVALUATIONS OF REALISM IN GENERATED VIDEOS?

A key challenge in evaluating generated videos is measuring realism without relying on reference videos. This is particularly relevant when training data are inaccessible or when sampling a large number of outputs is computationally prohibitive. Human evaluation has therefore become the gold standard, since people can naturally judge whether motion appears realistic and physically plausible.

| Model | Spearman (PC) |
|---|---|
| *Video Evaluation Models (fine-tuned)* | |
| VideoCon–Physics | 0.48 |
| VideoCon | 0.13 |
| VideoLlava | 0.08 |
| VideoScore | 0.17 |
| VIDEOPHY–2–AUTOEVAL | **0.76** |
| *TRAJAN Variants (no fine-tuning)* | |
| TRAJAN | 0.19 |
| TRAJAN+DINO | 0.40 |
| TRAJAN+3D | 0.50 |
| 3DSPA (ours) | **0.74** |

Table 3: Spearman rank coefficients on the VideoPhy-2 benchmark for physical commonsense (PC). Video Evaluation Models are fine-tuned vision-language models.

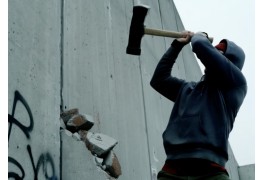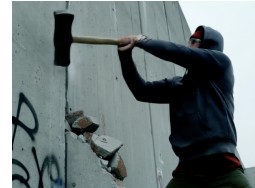
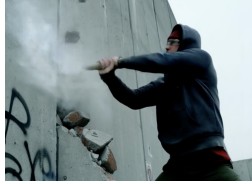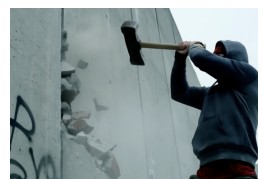

Figure 3: An example *unrealistic* video (Physical Commonsense score of 2/5) from VideoPhy-2 which 3DSPA scores poorly (Average 3D Jaccard of 6.95) but TRAJAN scores highly (Average 2D Jaccard of 60.9).

To ground our study, we draw on two datasets which include a large set of videos generated by a large collection of generative video models: VideoPhy-2 (Bansal et al. (2025)) and EvalCrafter (Skinner et al., 2023). VideoPhy-2 emphasizes action-centric videos and includes human annotations of physical commonsense and semantic adherence to the text prompt. EvalCrafter (Skinner et al. (2023)), evaluates video quality with a larger set of five metrics including motion quality, temporal consistency, and several prompt adherence measures.

**VideoPhy-2** We use the VideoPhy-2 benchmark (Bansal et al. (2025)) to assess how well 3DSPA performs as an automated realism metric relative to human judgments. This benchmark emphasizes two key aspects: *semantic adherence* (SA), which measures whether generated videos follow the intended action semantics, and *physical commonsense* (PC), which evaluates whether the motion and interactions in videos are consistent with intuitive physical rules. We are primarily interested in physical commonsense. Bansal et al. (2025) also provide an automated evaluation metric, VIDEOPHY-2 AutoEval, which is a vision-language model fine-tuned to predict a physical commonsense score on a subset of the generated video dataset.

We measure the automated metric quality by correlating model ratings and human ratings with the Spearman rank coefficient. Model ratings are calculated automatically as the Average Jaccard for each video – a proxy for the reconstruction error of the autoencoder.

As shown in Table 3, 3DSPA substantially outperforms 2D variants such as TRAJAN and TRAJAN+DINO in tracking human ratings of physical commonsense, and also provides a significant boost over the 3D baseline. The inclusion of both 3D structural cues and semantic DINO features enables stronger alignment with human assessments, where 3DSPA achieves the highest Spearman rank coefficient among TRAJAN variants. More remarkably, 3DSPA strongly outperforms most vision-language models (VideoCon, VideoScore, and VideoLlava) on this task, and even closely matches VIDEOPHY-2 AutoEval despite not being trained on the provided dataset.

Figure 3 shows an example video which highlights the difference between 3DSPA and TRAJAN in capturing physical commonsense. In this video of a man smashing a concrete wall with a hammer, the motions are

smooth but the hammer also partially disappears. TRAJAN assigns a high realism rating because it is only sensitive to the smooth motion. 3DSPA assigns the video a low score because it is additionally sensitive to the semantics: the hammer cannot just disappear. This underscores the importance of semantic information when evaluating physical realism.

**EvalCrafter** Similarly to VideoPhy-2, EvalCrafter (Skinner et al. (2023)) is a dataset consisting of generated videos from several frontier generative video models. For each video, a set of human annotators rated the **visual quality**, **text to video consistency**, **motion quality**, **temporal consistency**, and **subjective likeness**. Similarly to VideoPhy-2, we compute Spearman rank coefficients between human ratings and the Average Jaccard (AJ) for each of the TRAJAN variants and 3DSPA . Since many videos in EvalCrafter contain no motion (and therefore could not be assessed as physically realistic or not), we restricted evaluation to videos with medium to high motion, defined as the top 50% of videos ranked by change in 3D point track positions, yielding a test set of 1,849 videos. Table 4 clearly demonstrates that 3DSPA achieves the best performance; further highlighting that integrating both 3D structure and semantic DINO features provides the strongest predictor of a variety of human annotations for generated videos.

| Model | Visual Quality | T2V | Motion Quality | Consistency | Subjective Likeness |
|---|---|---|---|---|---|
| TRAJAN | 0.28 | 0.25 | 0.24 | 0.33 | 0.23 |
| TRAJAN+DINO | 0.31 | 0.44 | 0.41 | 0.39 | 0.46 |
| TRAJAN+3D | 0.45 | 0.55 | 0.49 | 0.48 | **0.63** |
| 3DSPA (ours) | **0.48** | **0.58** | **0.55** | **0.60** | 0.60 |

Table 4: Spearman rank coefficients between human annotations and automated AJ scores across different TRAJAN variants for the categories of visual quality, text-to-video similarity, motion quality, temporal consistency, and subjective likeness.

## 5 DISCUSSION AND CONCLUSION

We introduce the 3D Semantic Point Autoencoder (3DSPA ), a framework for evaluating video realism using semantic-aware 3D point trajectories. Across several experiments, we found that 3DSPA 's combination of semantic and 3D geometric information was crucial for (1) 3D point track reconstruction, (2) physical rule violation detection, and (3) matching various human annotations of generated videos, including motion quality and adherence to physical commonsense.

Through extensive ablation studies, we determined that semantic information is particularly crucial to determining whether a video is physically realistic – geometry alone is not enough. Perhaps most surprisingly, 3DSPA outperforms state-of-the-art vision-language models both in detecting synthetic physical rule violations such as solidity and immutability, and in tracking human physical commonsense judgements of generated videos.

Overall, 3DSPA offers a scalable alternative to human evaluation of video realism. We believe 3D point tracks naturally capture depth-aware motion, interactions, and occlusion, making them more effective than frame-based metrics for spotting subtle physics violations. In future work, we plan to make trajectories depend on past motion, enabling stronger tests of long-term dynamics and temporal realism, as well as investigate whether these metrics can be used to improve or regularize the training of generative video models.

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

## A  TRAINING SETUP & HYPERPARAMETERS

We train our model using AdamW (Loshchilov & Hutter, 2017) with a cosine learning rate schedule, preceded by a warmup of 10000 steps. The peak learning rate is set to $1 \times 10^{-4}$. Training is performed for 300 epochs with a batch size of 256. This extended training schedule, along with the larger batch size, allows the model to better stabilize its motion representation and improve generalization across diverse video scenarios.

As before, we supervise both position and occlusion prediction, applying a L1 loss on $(x_t, y_t)$ coordinates and a cross-entropy loss on the occlusion logit $o_t$. We maintain the weighting ratio of $5000 : 10^{-8}$, which

prioritizes motion fidelity while encouraging invariance to occlusion. We found that balancing these losses equally degraded correlation with human judgments of realism, consistent with our earlier observations.

To improve temporal localization of query points, we replace the naive linear up-projection operator with a strided-window upsampling operator. Specifically, each latent token $\phi_S^l$ is linearly up-projected and concatenated with a temporal window $[\rho_t : \rho_t + 128)$ along the channel axis. This encourages the decoder to attend to temporally relevant information for a given query point.

## A.1 HYPERPARAMETERS

Tables 5 and 6 provide the full set of hyperparameters for positional encoding, projection operators, and transformer modules. Compared to the original TRAJAN configuration, we increase the dimensionalities of the projection layers as we have increased the data and added additional parameters due to DINO and depth features, enabling richer multi-modal fusion of semantic and geometric cues.

| Component | Hyperparameter Value |
|---|---|
| Sinusoidal embedding (spatial + temporal + depth) | 32 frequencies |
| Track token projection dimensionality ($C$) | 384 |
| DINO feature projection dimensionality | 768 |
| Depth feature projection dimensionality | 256 |
| Compression dimensionality | 96 |
| Up-projection dimensionality | 1280 |
| Query point encoder dimensionality | 1280 |

Table 5: Positional encoding and projection operator hyperparameters for 3DSPA .

## B ADDITIONAL RESULTS ON INTPHYS2

**Dataset Structure.** The videos are further categorized by difficulty:

- **Easy (104 videos):** Simple environments with colorful geometric shapes.
- **Medium (400 videos):** Diverse backgrounds with textured shapes.
- **Hard (336 videos):** Realistic objects within cluttered, complex backgrounds.
- **Unknown (172 videos):** Mixed or ambiguous scenes.

We report results across the three difficulty variants to better capture how model performance scales with increasing visual and physical complexity. This breakdown provides motivation for our evaluation, as it disentangles robustness to simple synthetic settings from generalization to realistic and cluttered environments (see Figure 4). Notably, performance on the **hard** category is consistently the lowest in terms of Average Jaccard, highlighting the challenge of reconstructing tracks in realistic scenes with heavy clutter, occlusions, and object interactions.

We also give a qualitative example of how 3DSPA captures physical rule violations in Figure 5. Recall that the IntPhys2 dataset is constructed by pairing the same initial frames with either a realistic ending (real) or one which violates a physical rule (unreal). In Figure 5, the ball should interact with the ramp and fly through the air (top, real). In this case, the point tracks on the ball are well reconstructed. However, in the bottom video, where the ball just continues down the slope without flying through the air, the point tracks are poorly reconstructed.

| Transformer Name | Attention Type | QKV Size | Layers | Heads | MLP Size |
|---|---|---|---|---|---|
| Input 3D track transformer | SA | $96 \times 8$ | 3 | 8 | 1536 |
| Perceiver-style transformer | CA | $96 \times 8$ | 4 | 8 | 2048 |
| Up-projection latent transformer (decoder) | CA | $96 \times 8$ | 4 | 8 | 2048 |
| Track readout transformer | CA | $96 \times 8$ | 4 | 8 | 1536 |

Table 6: Transformer architecture hyperparameters for 3DSPA . SA = self-attention, CA = cross-attention.

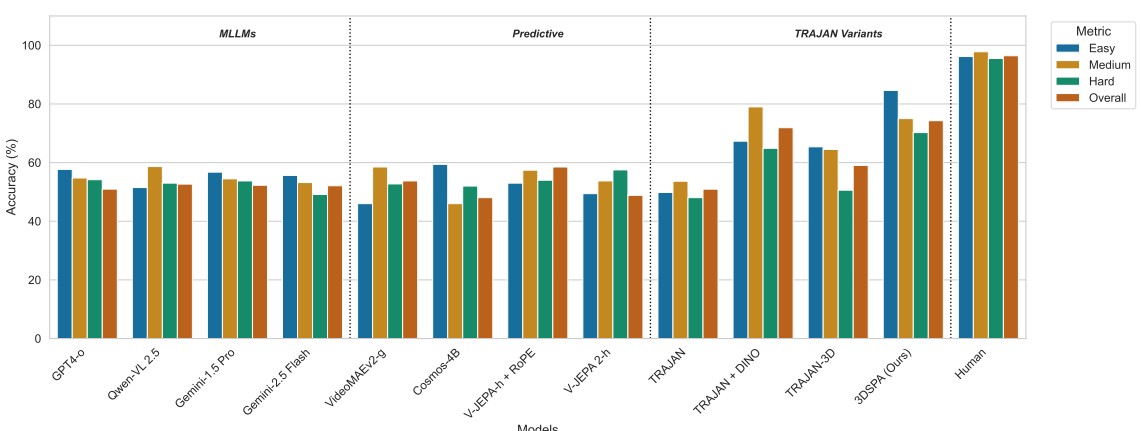

Figure 4: Performance comparison across models on the IntPhys2 benchmark for each of the *easy*, *medium* and *hard* categories.

## C    3DSPA SUCCESSES AND FAILURES ON 3D POINT TRACK RECONSTRUCTION

In this section, we give some examples of how well 3DSPA can reconstruct 3D point tracks on the TAP-Vid3D-Minival dataset. In Figure 6, we show successes of 3DSPA in reconstructing point tracks of both small objects and fast moving objects. However, 3DSPA struggles when depth cues are difficult to interpret (Figure 7). Since we rely on VideoDepthAnything (Chen et al., 2025a), when depth cues are ambiguous, we see poorer 3D tracking. However, the 2D tracks still look reasonable.

## D    VISUALIZATIONS COMPARING TRAJAN AND 3DSPA

In this section we provide visualizations comparing TRAJAN Allen et al. (2025) and 3DSPA . IN Figure 8 we present one case where the 3D nature of 3DSPA is critical to reconstructing the point tracks of a realistic video, and one where the semantic feature embeddings of 3DSPA are instead critical for predicting that an inanimate object cannot disappear arbitrarily.

## E    VISUALIZATIONS SHOWING 3DSPA 'S ABILITY TO CAPTURE GOOD AND BAD MOTION IN GENERATED VIDEOS

Here we show 3DSPA 's ability to capture realistic and unrealistic motion across a selection of generated videos. In Figure 9, we show three examples of videos from the Evalcrafter and VideoPhy2 datasets which

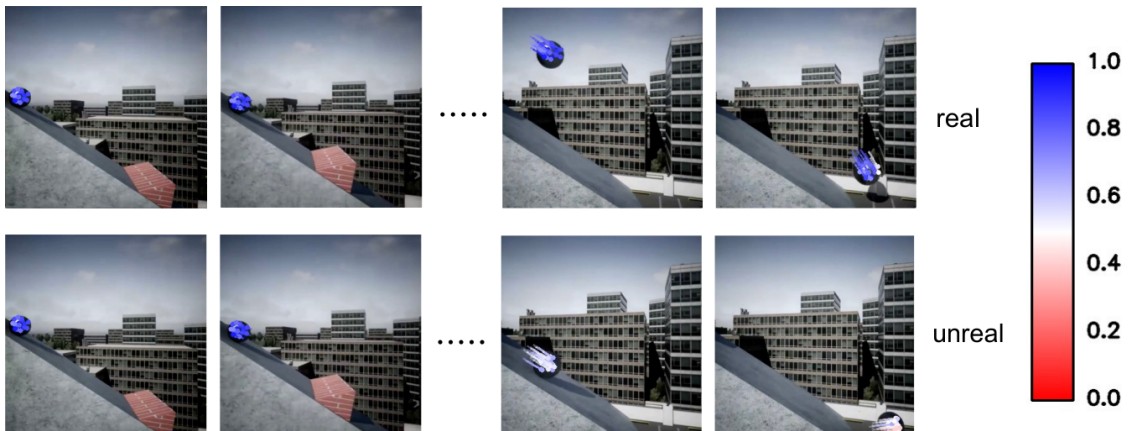

Figure 5: **Visualization of point tracks on IntPhys2 videos.** For each video, we project tracked points and color them using their Average Jaccard (AJ) consistency score (red = low / unreal, blue = high / real). Real, physically consistent videos (top) show smooth, coherent tracks with high AJ scores throughout the video, while unreal videos with physical rule violations (bottom) exhibit poorly reconstructed point trajectories.

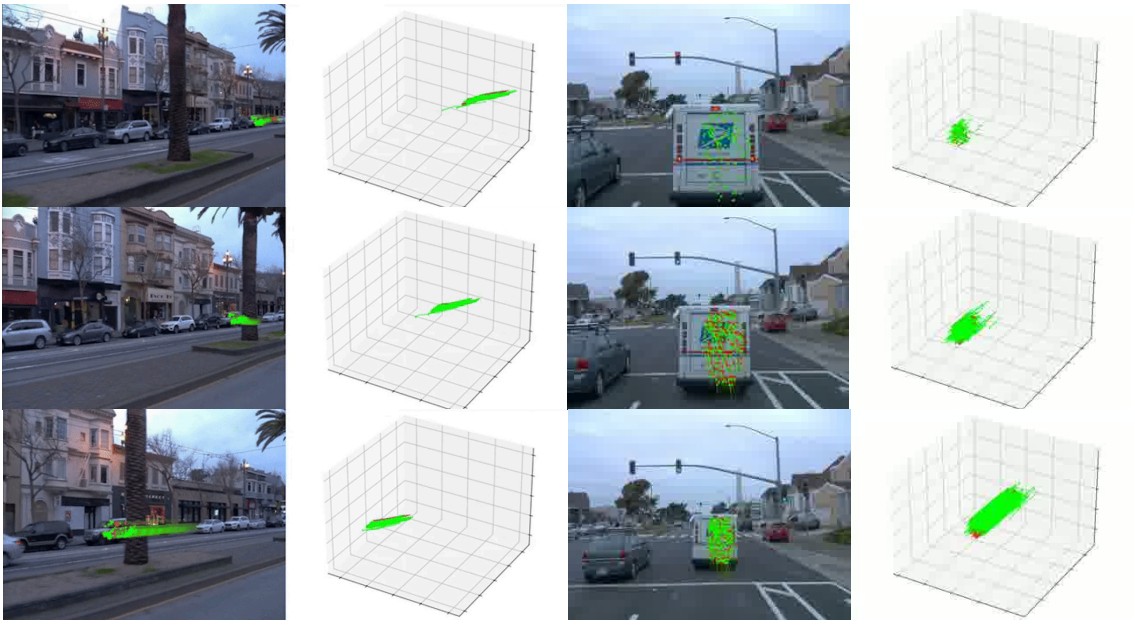

Red: ground truth, Green: prediction

Figure 6: **Positive examples where 3DSPA performs well on TAP-Vid3D-Minival.** Predicted 3D trajectories (green) align closely with ground truth (red) even for videos with small objects and high motion. These videos are from the DriveTrack dataset within TAP-Vid3D-Minival.

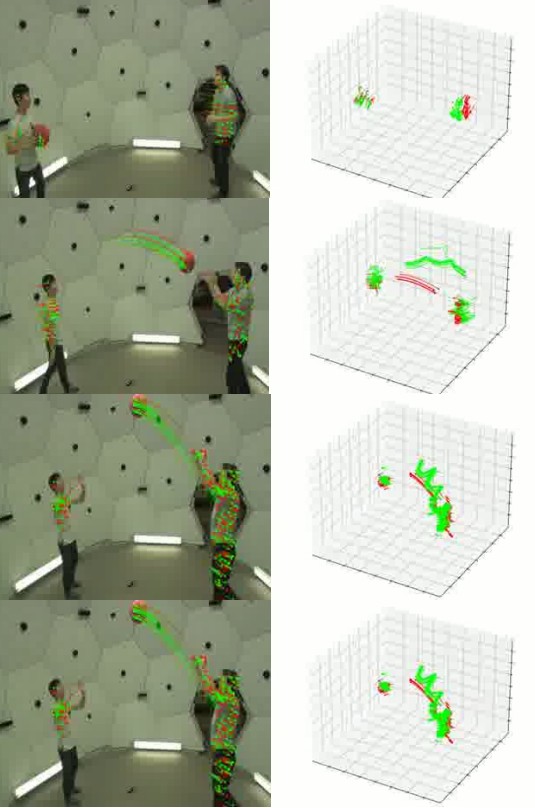

Red: ground truth, Green: prediction

Figure 7: **Failure cases of 3DSPA on TAP-Vid3D-Minival.** We visualize ground-truth (red) and predicted (green) query 3D point trajectories for challenging scenes. Errors typically arise in regions with complex geometry or depth ambiguities, giving high error in depth prediction, although the motion following is good. Corresponding 2D frames (left) and reconstructed 3D trajectories (right) highlight mismatches between predicted and true motion, which mainly arise from incorrectly estimated depth.

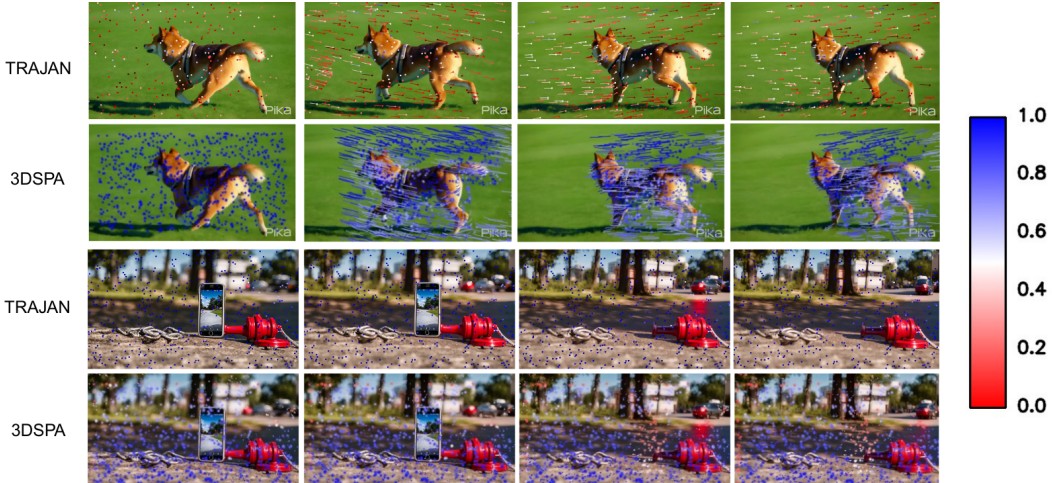

Figure 8: **TRAJAN vs. 3DSPA.** These are videos from the Evalcrafter dataset where compared to TRAJAN Allen et al. (2025), 3DSPA gives substantially more coherent and temporally stable point tracks, aligning better with human ratings for motion quality. The top video of a dog walking was rated 4.5/5. TRAJAN gives bad reconstructed point tracks across the video whereas 3DSPA gives good reconstructed points tracks, likely due to its ability to model the dog's legs in 3D. In the bottom video (rated 1.67/5 by humans), the phone disappears slowly, and only 3DSPA recognizes this as a problem due to its additional semantic feature embeddings relative to TRAJAN.

were all rated very highly for realistic motion and cover challenging situations for point tracking including transparency and high motion. In all cases, 3DSPA reconstructs the point tracks well, since the videos appear realistic.

In Figure 10, we instead show examples of videos that were rated *poorly* for motion realism. In these cases, objects are morphing shape or inappropriately disappearing. The tracks are poorly reconstructed exactly where the motion becomes unrealistic.

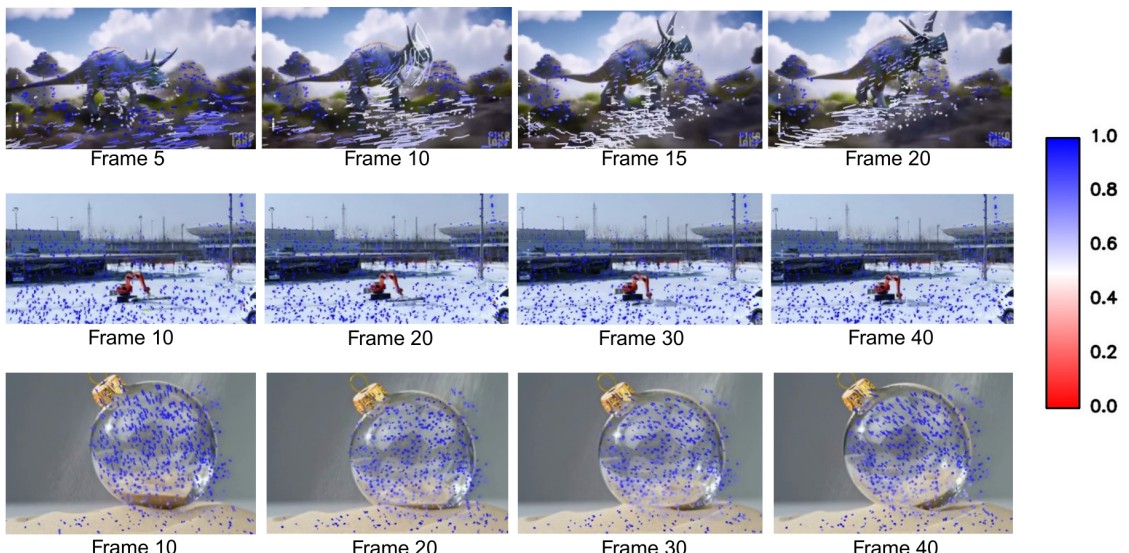

Figure 9: **Additional Visualization on Evalcrafter & VideoPhy2.** High-rated motion videos had 3DSPA produce coherent, stable point tracks. The top and middle videos are from Evalcrafter Skinner et al. (2023), while the bottom video is from Videophy2 Bansal et al. (2025).

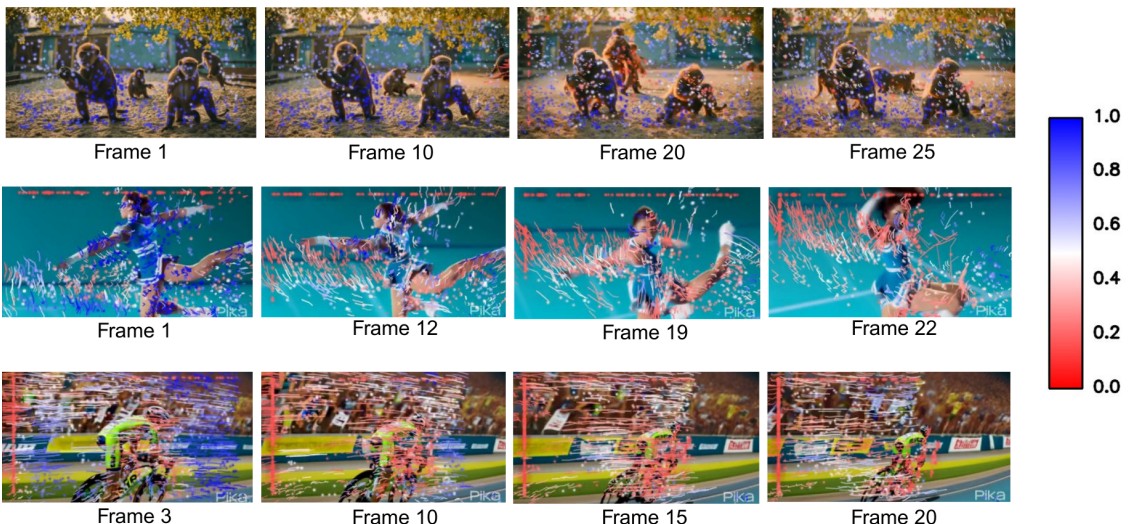

Figure 10: **Additional Visualization on Evalcrafter & Videophy2.** Low-rated motion videos lead to lower AJ scores and unstable point tracks from 3DSPA , reflecting difficulty in reconstructing unreliable motion. The top videos are from Evalcrafter Skinner et al. (2023), middle and bottom are from Videophy2 Bansal et al. (2025).

