# OpenReview forum: "3DSPA: A 3D Semantic Point Autoencoder for Evaluating Video Realism"
_ICLR.cc/2026/Conference — ICLR 2026 Conference Desk Rejected Submission_

### Official Review · Reviewer_QxzU · 2025-10-19

**Soundness:** 3
**Presentation:** 3
**Contribution:** 3
**Rating:** 8
**Confidence:** 4

**Summary:**

This paper proposes using 3D point cloud trajectory reconstruction as a criterion for evaluating the realism of video generation models, representing the first genuine attempt to address physical and motion inaccuracies in generated videos from a 3D perspective, which I believe is the proper way. By integrating both 3D and 2D semantic information, the authors construct a robust input representation. The method is trained on both synthetic and real-world datasets and is validated on TAPVid-3D, EvalCrafter, and VideoPhy2, demonstrating its effectiveness

**Strengths:**

1. This paper firstly proposes the way to evaluate the realism of generated videos in a 3D perspective, which is a promising way for future works.
2. The result shows that 3DSPA works accurately both in real and generated videos.
3. The whole paper is well writen and easy to understand.

**Weaknesses:**

1. The authors did not provide a clear rationale in the paper for the necessity of the supported point track input; its effectiveness is only demonstrated through ablation experiments. It would be better if the paper included more discussion and justification for this component.
2. The way for selecting the initial point is unclear, if utilizing uniform selection strategy, how to ensure the point will fall into the entity with motion or fast motion (the scene/entity changes several time in a video)
3. The time costing compared with 2d vlm-based methods, which should be reported as it is important for a evaluation methods. (Note that what I would like to know is the time required to reproduce the results presented in the paper. )

**Questions:**

See Weaknesses.

---

> ### Author Response · Authors · 2025-11-23
> **Response**
>
> Thank you for your valuable feedback. Below, we respond to each of the issues you raised. Some of our responses include additional experiments or analyses that will be incorporated into the further revisions. We believe the following clarifications address your concerns and questions and can be accommodated within the one-page addition permitted in the camera-ready version.
>
> __W1: Necessity of Supported Point Tracks__
>
> The support point tracks serve two critical purposes in 3DSPA's design:
>
> -> The support tracks provide the encoder with a dense sampling of motion across the video, allowing it to capture global motion patterns, scene dynamics, and physical regularities. These latents are powered by semantic and depth cues and motion information in the point tracks in the enlarged space. This compressed representation in the motion latent space is what enables the model to reason about whether query track predictions are physically plausible.
>
> ->By forcing the model to compress diverse trajectories into a fixed-size latent and then reconstruct held-out query tracks, we create an information bottleneck that encourages the model to learn fundamental physical and semantic regularities rather than memorizing individual trajectories. Unrealistic videos that violate physical laws will produce trajectories that cannot be well-reconstructed from this bottleneck, resulting in high reconstruction error (low AJ).
>
> __W2: Point Selection Strategy__
>
> You raise an important question about how we select initial points.
>
> Our point selection strategy follows the CoTracker3 approach for dense tracking: we use a uniform grid sampling strategy across the first frame.
>
> __Ensuring coverage of moving objects:__ While this sampling across space and time doesn't explicitly target moving regions, in practice it provides sufficient coverage because:
>
> 1. In our evaluation dataset of Evalcrafter and VideoPhy, we focus on action-centric content where we filter out the low motion videos. Most videos contain substantial motion.
> 2. We sample a large number of points (typically 512-1024 per video), ensuring high probability of capturing moving objects
> 3. Our occlusion-aware attention mechanism allows the model to focus on visible, trackable points while ignoring occluded or unreliable tracks.
>
> __W3: Time Cost Comparison__
>
> We record the inference time using an A100-40GB GPU compared to several baselines below (with all but VideoScore being based on VLMs). We find that 3DSPA (and TRAJAN2D) are both substantially faster than the VLM models. Training 3DSPA took 1 day on 8 A100-40GB GPUs, representing a fairly reasonable computational cost.
>
> | Model | Inference time |
> |---|---|
> | VideoCon–Physics | 8.13s |
> | VideoCon | 6.21s |
> | VideoLLaVA | 13.52s|
> | VideoScore | 4.89s |
> | VIDEOPHY–2–AUTOEVAL | 10.67s |
> | Trajan2D | 2.18s|
> | 3DSPA |5.61s|

---

### Official Review · Reviewer_Yrng · 2025-10-31

**Soundness:** 2
**Presentation:** 2
**Contribution:** 2
**Rating:** 6
**Confidence:** 2

**Summary:**

The paper introduces 3DSPA (3D Semantic Point Autoencoder), an automated framework for evaluating the realism of generated videos, addressing limitations of existing methods in assessing video realism, temporal, and physical plausibility. By integrating 3D point trajectories, depth cues, and DINO semantic features into a unified representation, 3DSPA enables robust assessment of video realism, temporal consistency, and physical plausibility without requiring a reference video. Its primary contribution lies in its ability to reliably identify videos that violate physical laws, its high sensitivity to motion artifacts, and its closer alignment with human judgments of video quality and realism, even outperforming state-of-the-art vision-language models in detecting physical inconsistencies. The research highlights that the combination of semantic and 3D geometric information is crucial for judging physical realism, and that 3DSPA offers a scalable and effective alternative to the labor-intensive human evaluation of video realism.

**Strengths:**

Novel Framework: 3DSPA integrates 3D point trajectories, depth cues, and DINO semantic features into a unified representation, allowing for robust assessments of video realism, temporal consistency, and physical plausibility without requiring a reference video.

Enhanced Realism Detection: It reliably identifies videos that violate physical laws, is highly sensitive to motion artifacts, and aligns more closely with human judgments of video quality and realism compared to existing methods.

Superior Performance: Through extensive experiments, 3DSPA demonstrates better performance in 3D point track reconstruction, physical rule violation detection, and matching human annotations of realism, even outperforming state-of-the-art vision-language models in detecting physical inconsistencies. The authors highlight that the combination of semantic and 3D geometric information is crucial, with semantic information being particularly important for judging physical realism.

Scalable Alternative: 3DSPA offers a scalable and effective alternative to the labor-intensive human evaluation of video realism.

**Weaknesses:**

1. Representation weakness: Table 1 is out of page, and the paper is less than 9 pages in length.
2. The difference between 3DSPA and TRAJAN+DINO/+3D need more disscusions.
3. Lacking of ablations: There is no ablation studies in paper and supp. Only comparison with TRAJAN variants are involved. Ablations of different components in 3DSPA are expected.
4. More visualization results are expected, including 3D point tracks reconstructed by 3DSPA for more videos, and more unrealistic videos which 3DSPA scores poorly but TRAJAN scores highly.

**Questions:**

See weakness.

---

> ### Author Response · Authors · 2025-11-23
> **Response**
>
> Thank you for your valuable feedback. Below, we respond to each of the issues you raised. Some of our responses include additional experiments or analyses that will be incorporated into further revisions. We believe the following clarifications address your concerns and questions and can be accommodated within the one-page addition permitted in the camera-ready version.
>
> __W1: Table Formatting__
>
> We apologize for the formatting issue. We have reformatted Table 1 in the revision to fit within the page boundaries.
>
> __W2: Difference Between 3DSPA and TRAJAN+DINO/+3D__
>
> Since 3DSPA can be viewed as an extension of TRAJAN augmented with depth and semantic cues, we introduce two intermediate baselines, each adding one component at a time, to illustrate the incremental improvements as we transition from TRAJAN to the full 3DSPA model. These are treated as baselines in the text, but one can clearly argue that these are the ablated versions of 3DSPA. We talk about each variation in Section 4.2 and give more information below for convenience.
>
> The key differences are:
>
>
>
> * __TRAJAN+DINO:__ Adds semantic features (DINOv2) to 2D trajectories but lacks depth information. It cannot reason about 3D spatial relationships, occlusion in 3D space, or depth-aware physical plausibility.
> * __TRAJAN+3D:__ Extends trajectories to 3D using depth but lacks semantic context. It can capture geometric motion but cannot leverage object identity or semantic information.
> * __3DSPA:__ Jointly integrates both 3D geometry AND semantic features. This allows reasoning about both "what objects are" (semantics) and "how they move in 3D space" (geometry), which is crucial for judging physical realism.
>
> As shown in Table 2 (IntPhys2), TRAJAN+DINO performs comparably to 3DSPA, suggesting that semantics are particularly important for physical reasoning. However, Table 4 (EvalCrafter) shows 3DSPA consistently outperforms both ablations, demonstrating that the combination of 3D+semantics provides complementary benefits that neither alone achieves.
>
> __W3: Lacking Ablations__
>
> As mentioned in the argument for W2, we view each intermediate model as ablations.
>
> __W4: More Visualizations__
>
> We agree that more visualizations will help the readers. We have added the following visualizations to the appendix/main paper in the revised version.
>
> * 3D point track reconstructions for 4 diverse videos spanning different categories and types of motion in Section C
> * Visualizations of point tracks for unreal and real videos for IntPhys2 in Section B
> * Direct comparisons of TRAJAN and 3DSPA demonstrating where 3DSPA correctly predicts human ratings but TRAJAN does not in Section D
> * Further examples of where 3DSPA predicts both good and bad motion in generated videos in Section E

---

### Official Review · Reviewer_Fjpp · 2025-11-01

**Soundness:** 3
**Presentation:** 3
**Contribution:** 3
**Rating:** 6
**Confidence:** 2

**Summary:**

This paper proposes 3DSPA, a novel framework for evaluating video realism. It encodes 3D point trajectories coupled with depth cues and DINOv2 semantic embeddings to jointly capture motion dynamics and physical plausibility. Experiments on diverse datasets show that 3DSPA effectively captures motion and physical information and correlates closely with human realism judgments.

**Strengths:**

1. Paper is well written and organized.

2. Novel 3D semantic-physical representation: Proposes a 3D point autoencoder that jointly encodes geometric motion and semantic features, allowing more comprehensive realism assessment..

3.  Strong empirical validation and interpretability: Experiments valid  3DSPA is capable of reconstructing 3D point tracks and detect physical rule violations. Extensive results show consistent superiority over previous evaluators.

**Weaknesses:**

1. Reliance on additional models: The 3d points are obtained with CoTracker3 and Video Depth Anything. These models are strong in point tracking and metric depth estimation, yet still have limitations, for example, cotracker3 may fail with small objects in the video and VDA may face issues with sharp lines. Also, dramatic motions or camera movements remain problematic in geometric models. Will these possible artifacts or errors of geometric prediction affect the results of 3DSPA? Look forward to authors' analysis or discussion on these cases.
2. Inference efficiency: With the introduction of multiple geometric and semantic model, comparison of compute cost with existing baselines should be provided.
3. Failure case analysis: Include some failure case on motion tracking would benefit this paper. I am curious where the failure come from, hallucination from semantic model or geometric model.

**Questions:**

Please refer to weakness.

Additional questions

4.  generalization: The performance of 3DSPA is closely related to training data. How 3DSPA perform on more diverse videos (e.g. robotics / autonomous driving / 3d scene / games ...) What is the cost and performance if fine-tune 3DSPA to do evaluation on specific domains?

---

> ### Author Response · Authors · 2025-11-23
> **Response 1/2**
>
> Thank you for your valuable feedback. Below, we respond to each of the issues you raised. Some of our responses include additional experiments or analyses that will be incorporated into the further revisions. We believe the following clarifications address your concerns and questions and can be accommodated within the one-page addition permitted in the camera-ready version.
>
> __W1: Reliance on Additional Models__
>
> We appreciate your concern about the reliance on CoTracker3 and Video Depth Anything (VDA). You are correct that these models have limitations, and we agree that analyzing their failure modes is important.
>
> While the weaknesses of the models are rightfully pointed out, our experimental results suggest that 3DSPA is reasonably robust to different environments and these artifacts do not have a huge effect on the performance. In Table 1, we show that 3DSPA achieves competitive performance on the TAPVid-3D minival set. Even when we finetuned CoTracker3 on the TapVid-3D main set and incorporated ZoeDepth (an equally strong model as DepthAnythingv2 (on which VDA is built on)), 3DSPA performs better in terms of the AJ metrics which shows the accuracy of how closely the tracked points match the ground truth.
>
> __Dramatic motions and camera movements:__ You raise an excellent point about dramatic motions or fast camera movements potentially causing issues in geometric models. We note that our training data includes diverse motion patterns and camera movements, which helps 3DSPA learn to handle such scenarios. Additionally, on the IntPhys2 benchmark (Table 2), which includes both static and moving cameras, 3DSPA maintains strong performance across both conditions (for example, 76.92% vs 75.00% on Permanence).
>
> We now include an analysis section with several visualizations in the appendix (Section C). We would like to point out that 3DSPA is not intended to perfectly reconstruct all point tracks. It is intended to reconstruct point tracks when those point tracks come from realistic motion. We *expect* it to fail when the point tracks are coming from unrealistic motion. Nevertheless, for videos from the TAPVid-3D-Minival dataset, which consists entirely of real videos, we find that 3DSPA can successfully reconstruct fast-moving and small objects.
>
> __W2: Inference Efficiency__
>
> At inference time, 3DSPA requires running CoTracker3 (for 2D tracking), VDA (for depth estimation), DINOv2 (for semantic features), and implementation of our autoencoder. We record the inference time using an A100-40GB GPU compared to several baselines below. We report the average inference time using 50 frames across 50 different videos.
> | Model | Inference time |
> |---|---|
> | VideoCon–Physics | 8.13s |
> | VideoCon | 6.21s |
> | VideoLLaVA | 13.52s|
> | VideoScore | 4.89s |
> | VIDEOPHY–2–AUTOEVAL | 10.67s |
> | Trajan2D | 2.18s|
> | 3DSPA |5.61s|
>
> We find that while 3DSPA is about 2x slower than TRAJAN, it is significantly faster than the various VLM baselines that are commonly used to evaluate physical commonsense in videos.
>
> We are happy to add these results to the paper or Appendix, whichever the reviewer feels would be most appropriate.
>
> __W3: Failure Cases Analysis__
>
> We now have a dedicated section of the appendix (Section C, see revision) showing both successes and failures of 3DSPA on the TAPVid-3D-Minival dataset. In our analyses, we found that 3DSPA can reconstruct fast-moving objects of both small and large sizes, but fails when the depth estimation from VideoDepthAnything fails. Interestingly, in these cases, the 2D point tracks still look reasonably accurate, but the 3D estimation is significantly off.

---

> ### Author Response · Authors · 2025-11-23
> **Response 2/2**
>
> __W4: Generalization to Diverse Domains__
>
> Our current evaluation spans diverse visual content which differs significantly from training data. TAPVid-3D (our training data) includes real-world egocentric rooms (Aria), driving (DriveTrack), and controlled handheld videos. EvalCrafter and VideoPhy-2 (our evaluation data) cover a wide range of generated content from multiple generative video models, which may look nothing like the TAPVid-3D dataset. Intphys2 (another evaluation dataset) contains simulated videos with simple game mechanics which look nothing like the videos in TAPVid-3D. As a result, we think our current analyses *already indicate strong generalization* of our model. However, we acknowledge that specialized domains like robotics simulation may have different motion statistics. 3DSPA could be very useful for robotics domains as well, but since our focus is on evaluating generated video content, we believe this falls outside the scope of this paper.
>
> __Cost analysis on finetuning:__
>
> Fine-tuning 3DSPA ideally requires labeled 3D trajectories with occlusions. If such annotations are unavailable but we have diverse videos (outside Kubric, PointOdyssey and Tapvid3d), we can generate pseudo-labels using a 3d point tracker and then run experiments similar to our training procedure. We estimate that fine-tuning on 5K domain-specific videos (achievable with pseudo-labeling) for 100 epochs would provide substantial domain adaptation at moderate computational cost of 1 day on 8 A100-40GB GPUs, since this was approximately the cost to train on the TAPVid-3D dataset (4569 videos of 50-200 frames each).

---

### Author Response · Authors · 2025-12-02
**Summary of reviews & response**

Here we summarize the reviews and our response for the new AC.

## Summary

In this work, we introduce 3DSPA, a 3D spatiotemporal point autoencoder that unifies 3D point trajectories, depth information, and DINO-based semantic cues into a single representation for video evaluation. By jointly modeling how objects move, how they occupy 3D space, and what they represent semantically, 3DSPA provides a holistic understanding of scene dynamics. This enables robust assessments of realism, temporal coherence, and physical plausibility.

Across diverse datasets, 3DSPA consistently detects violations of physical laws, exhibits heightened sensitivity to motion and geometry artifacts, and correlates more strongly with human judgments of video quality.

In the following global response, we address all the key concerns raised by the reviewers and further demonstrate the feasibility and effectiveness of 3DSPA.


## __Q1: Dependence on CoTracker3 and Video Depth Anything (VDA); robustness to geometric artifacts _(Reviewer Fjpp Weakness 1; Reviewer QxzU Weakness 1)___

__A1:__ CoTracker3 and VDA have known limitations (e.g., small objects, sharp lines, fast camera motion). However, our experiments show that 3DSPA is robust to these artifacts and performs well across diverse environments.

* In Table 1, 3DSPA achieves competitive results even when CoTracker3 is fine-tuned on TAPVid-3D.
* On IntPhys2 (Table 2), which includes both static and moving cameras, 3DSPA maintains strong physical-reasoning performance (for example, 76.92% (static camera) vs 75.00% (moving camera) on Permanence).
* Importantly, 3DSPA does not need to reconstruct every track perfectly, it only needs to reconstruct tracks arising from _realistic motion_. Unrealistic motion naturally leads to higher reconstruction error, which is desirable behavior.

We added a dedicated visualization section (Appendix C) showing both successes and failures as requested by the reviewer. We find that failures arise primarily from depth misestimation, not from the semantic model or the autoencoder.

## __Q2: Inference Efficiency and Compute Cost _(Reviewer Fjpp Weakness 2; Reviewer QxzU Weakness 3)___

__A2:__ 3DSPA’s pipeline includes CoTracker3 (2D tracking), VDA (depth), DINOv2 (semantic features), and the autoencoder. On an A100-40GB GPU, we measure average inference time over 50 videos × 50 frames:

| Model | Inference time |
|---|---|
| VideoCon–Physics | 8.13s |
| VideoCon | 6.21s |
| VideoLLaVA | 13.52s|
| VideoScore | 4.89s |
| VIDEOPHY–2–AUTOEVAL | 10.67s |
| Trajan2D | 2.18s|
| 3DSPA |5.61s|

Thus, 3DSPA is substantially faster than VLM-based evaluators while offering stronger physical realism assessment. \
Training cost is modest: ~1 day on 8×A100-40GB GPUs for TAPVid-3D training scale (approximately 4700 videos).


## __Q3: Failure Case Analysis _(Reviewer Fjpp Weakness 3)___

__A3:__ We now include several failure visualizations in Appendix C.

3DSPA reliably reconstructs fast-moving small and large objects in real videos. Failures occur primarily when depth estimation fails, even when 2D tracking remains accurate. These errors propagate to the 3D reconstruction and are correctly scored as unrealistic in generated videos.


## __Q4 Generalization to More Diverse Domains _(Reviewer Fjpp Additional Question; Reviewer QxzU Requests; Reviewer Yrng Weakness 3)___

__A4:__ Our evaluation already spans a wide range of distributions: egocentric real videos, driving scenes, simulated physics environments, and diverse generative videos. Results on EvalCrafter and Videophy2 (generated videos across 10 generative video models) and IntPhys2 (simulation) show strong cross-domain generalization.

Robotics or specialized simulations may feature different motion statistics; adapting 3DSPA may require domain-specific trajectories. If explicit annotations are missing, pseudo-labels can be generated with a 3D tracker. Fine-tuning on 5K domain videos for 100 epochs would take ~1 day (8×A100), similar to our TAPVid-3D training budget.


## __Q5: Differences between 3DSPA, TRAJAN+DINO, and TRAJAN+3D and Ablation Studies _(Reviewer Yrng Weakness 2, Weakness 3)___

__A5:__ The two intermediate baselines (TRAJAN+DINO and TRAJAN+3D) already act as ablations. We expanded the discussion in Sec. 4.2 and in the appendix to clarify this. Together, the results demonstrate the importance of both semantic and depth cues.

* TRAJAN+DINO: Semantic features added to 2D trajectories; cannot reason about 3D geometry or depth-aware plausibility.
* TRAJAN+3D: 3D trajectories only; lacks semantic understanding, object identity, and context.
* 3DSPA: Integrates 3D geometry + semantics, enabling reasoning about both _what_ objects are and _how_ they should move in 3D space.

---

> ### Author Response · Authors · 2025-12-02
> **Global response 2/2**
>
> ## __Q6: More Visualizations _(Reviewer Yrng Weakness 4)___
>
> __A6:__ We added:
>
> * 3D reconstructions for multiple real and generated videos (Appendix C)
> * Physical violation examples where 3DSPA succeeds but TRAJAN fails (Appendix D)
> * Unrealistic videos where 3DSPA scores low but semantics-only baselines score high (Appendix B/E)
>
> These provide a comprehensive qualitative understanding of 3DSPA’s behavior.
>
> ## __Q7: Point Selection Strategy Clarification _(Reviewer QxzU Weakness 2)___
>
> __A7__: Our point selection follows uniform grid sampling, consistent with CoTracker3. Despite uniform sampling:
>
> * We sample 512–1024 points, giving a high probability of capturing motion.
> * Our datasets (EvalCrafter, VideoPhy-2) are action-centric, ensuring the points fall on dynamic objects.
> * The occlusion-aware attention mechanism naturally ignores unreliable points.
>
> Thus, the strategy is simple yet robust.

---

### Note · Program_Chairs · 2026-01-17
**Submission Desk Rejected by Program Chairs**

The following references in this submission do not refer to real documents and/or have major errors in bibliographic information:

 James Skinner et al. Evalcrafter: Benchmarking video generation models with comprehensive evaluation. In NeurIPS Datasets and Benchmarks, 2023.
Chen Sun Wu et al. Tcc: Time-contrastive networks for self-supervised video representation learning. In CVPR, 2021.